# Secondary Structure in Amyloids in Relation to Their Wild Type Forms

**DOI:** 10.3390/ijms24010154

**Published:** 2022-12-21

**Authors:** Irena Roterman, Katarzyna Stapor, Leszek Konieczny

**Affiliations:** 1Department of Bioinformatics and Telemedicine, Jagiellonian University—Medical College, Medyczna 7, 30-688 Krakow, Poland; 2Department of Applied Informatics, Silesian University of Technology, Akademicka 2A, 44-100 Gliwice, Poland; 3Chair of Medical Biochemistry, Jagiellonian University—Medical College, Kopernika 7, 31-034 Krakow, Poland

**Keywords:** secondary structure, hydrogen bonds, amyloids, alpha-synuclein, transthyretin, Abeta, light chain of IgG, misfolding

## Abstract

The amyloid structures and their wild type forms, available in the PDB database, provide the basis for comparative analyses. Globular proteins are characterised by a 3D spatial structure, while a chain in any amyloid fibril has a 2D structure. Another difference lies in the structuring of the hydrogen bond network. Amyloid forms theoretically engage all the NH and C=O groups of the peptide bonds in a chain with two hydrogen bonds each. In addition, the hydrogen bond network is highly ordered—as perpendicular to the plane of the chain. The β-structure segments provide the hydrogen bond system with an anti-parallel system. The folds appearing in the rectilinear propagation of the segment with the β-structure are caused by just by one of the residues in the sequence—residues with a Rα-helical or Lα-helical conformation. The antiparallel system of the hydrogen bonds in the β-structure sections at the site of the amino acid with a Rα- or Lα-helical conformation changes into a parallel system locally. This system also ensures that the involvement of the C=O and H-N groups in the construction of the interchain hydrogen bond, while maintaining a perpendicular orientation towards the plane of the chain. Conformational analysis at the level of the Phi and Psi angles indicates the presence of the conditions for the structures observed in the amyloids. The specificity of amyloid structures with the dominant conformation expressed as |Psi| = |Phi| reveals the system of organisation present in amyloid fibrils. The Phi, Psi angles, as present in this particular structure, transformed to form |Psi| = |Phi| appear to be ordered co-linearly. Therefore, the calculation of the correlation coefficient may express the distribution around this idealised localisation on the Ramachandran map. Additionally, when the outstanding points are eliminated, the part of amyloid chain can be classified as fulfilling the defined conditions. In addition, the presentation of the chain structure using geometric parameters, V-angle—the angle between the planes of the adjacent peptide bonds (angle versus the virtual axis Cα-Cα) and the radius of the curvature R, depending on the size of the angle V, allows for a quantitative assessment of changes during amyloid transformation.

## 1. Introduction

Proteins in the amyloid form have been the subject of analysis due to their medical importance, as they disturb the functioning of the nervous system [1,2,3,4]. Diseases caused by the presence of pathological structural protein forms are collectively referred to as neuro-degenerative diseases [5,6]. The search for drugs to treat these diseases is aimed at blocking the propagation of fibrils, which disrupt the proper functioning of cells, including in particular cells of the nervous system [7]. However, the activity of other organs, such as the kidneys or heart, can also be damaged by the amyloid process [8]. The main area of research from the perspective of treatment is gene therapy; in particular, the use of stem cells [9,10].

Regardless of the advances made in therapeutic techniques, amyloid transformation is, in itself, an independent research object, the goal of which is to identify the phenomenon at the molecular level, especially in the context of the protein-folding problem [11,12,13]. A non-chemical technique for obtaining the amyloid forms of numerous proteins is protein shaking, which results in the appearance of amyloid forms after a sufficiently long shaking period. This observation is reinforced by studies on the influence of various types of interphase on the course of reactions, including air-water and hydrophobic surface-water interphases [14,15,16,17,18,19,20,21]. What is surprising in the first stages of these studies is the possibility also of the transformation into the amyloid form—and thus one rich in β-structure—of proteins with a considerable presence of a helical structure [22], including Alpha-synuclein [23,24]. Thanks to the extensive literature on the experimental research, there is an increasing number of amyloid structures accessible in PDB resources, which thus enables an analysis of the structure of amyloid fibrils, as in [25,26,27].

Experiments conducted in different environmental conditions point to the significant influence exerted by the environment, and thus the influence of external conditions [14,18,19,20,21,28].

A distinguishing structural feature of amyloid fibrils is the flatness of each chain—their fibril component. This arrangement is achieved through the appropriate sets of Phi and Psi angles in the amino acids in chain. The β-structure segments constitute a specific system realised by angles with the Psi = −Phi relation. Such a set of conformations along the appropriate section results in the C=O groups of peptide bonds oriented in a direction perpendicular to the plane of the chain, while the N-H groups of the peptide bonds also have a perpendicular orientation, but in the opposite direction. As a result, each amino acid is involved in the construction of two hydrogen bonds. The flat structure of the chain, with the presence of such a structure, can engage each peptide bond in building two hydrogen bonds, thereby helping to stabilise the fibril system. A polypeptide chain with Psi = −Phi angles can propagate indefinitely. The turns in the chain are derived from a single amino acid in the chain with an R- or L-α-helical conformation (see explanation in Section 2.1). This conformation will also orientate the C=O and H-N groups in a direction perpendicular to the plane of the chain, thus disrupting the order of the hydrogen bond. While in the case of the β-structure, successive hydrogen bonds remain in an anti-parallel system, the presence of a single amino acid with the Rα or Lα conformation of the helix locally creates a parallel bond system while maintaining the flatness of the structure. The presence of two residues with an Rα conformation or two amino acids with an Lα conformation would disturb the planar structure, bringing the chain orientation out of the plane. On the other hand, the presence of two amino acids with alternating Rα and Lα-helical conformations results in the appearance of a zigzag structure, but the flatness of the chain is maintained. The system of the hydrogen bond turns becomes locally parallel to the anti-parallel system in the β-sheet. The phrases occurring in specific positions in the fibril appear in each successive chain in the same place. Therefore, all available amyloid structures represent a parallel interchain system. This description has already been applied in the interpretation of a transthyretin amyloid [29].

The schematic presentation of the assumptions of the model of the idealised structure for amyloid fibril expressed by the relationship |Psi| = |Phi| is shown in Figure 1.

A simplified description will be given here, which will make it possible to track and interpret the results given later in the work.

Traditionally, the polypeptide chain structure has been described in terms of Phi and Psi angles expressing a specific conformation. The rotation on the Cα-N bonds—referred to as the rotation of Phi- and Cα-C’ (C’-C atom of the C=O group)—determines the value of the Psi angle. However, the effect of these rotations can be expressed using another parameter that also gives the effect of the rotations performed. This parameter is the V angle, which expresses the turn between two adjacent peptide bond planes as measured by the extent of the rotation of the peptide bond plane versus the rotation on the virtual Cα-Cα bond. The value of this angle varies from 0 deg for various forms of a helix up to the value V = 180 deg for a β-structure. All intermediate structures can be defined as corresponding values of V in the range 0 < V < 180 deg.

Changing the V angle (changes in the Phi and Psi angles) results in a different radius of curvature—from a helix with a known, small radius of curvature up to a theoretically infinite value for β-structure. All intermediate structures (for intermediate values of the V angle) assume forms with the value of the radius of the curvature R lying within this range.

The relationship determined for energetically acceptable structures on the Ramachandran map is given in the Section 4). The graph presented there shows the changes in the size of the radius of curvature (expressed on the Ln (R) scale) from the V-angle value.

The rectilinear propagation of the β-structure with an idealised system of hydrogen bonds generated by the C=O and H-N groups of peptide bonds involving all these groups within the backbone is obtained when angles Phi and Psi satisfy the condition Psi = −Phi. The present turns in the chain structures within the amyloid fibrils are achieved by single helical Rα and/or Lα conformations. Single conformations of this type do not disturb the planar system. The alternation of Rα and Lα does not disturb the flat structure of the chain. The appearance of two adjacent amino acids with one of these conformations results in a spatiality not observed in amyloid fibril structures.

The position of the points representing the Psi = −Phi conformation and the Rα or Lα conformation bears certain characteristics. The designated position of the angle values |Phi| and |Psi| reveals the location of these points on one common straight line. Therefore, each protein (in both native and amyloid forms) discussed in this paper is characterised by the value of the correlation coefficient CC. A distinction is made between CC-0 expressing order in the complete chain and the CC-F status of points after eliminating outstanding points.

The availability of amyloid structures together with their WT versions makes possible a comparative analysis. For this purpose, the following amyloid structures were used: transthyretin, α-synuclein and the V-domain of IgG. Additionally (despite no access to the WT structure), amyloids from the ABeta group were also discussed.

Proposals for an analysis of the polypeptide structures are presented in [28]. The basics of such an interpretation of the structure are given in detail in the Section 4 and in [30].

## 2. Results

The analysis concerns the comparison of proteins whose structures were available at the time of preparation of this paper in the PDB in both the native and amyloid forms. The distinction between the two forms is expressed as “amyloid” for the amyloid form and “native protein” for the form that the protein represents that exhibits biological activity.

### 2.1. Models of an Idealised Amyloid Form of a Polypeptide Chain

The idealised system for generating a planar structure constructed mainly from a β-structure combined with single (one amino acid in the sequence) residues with a Rα or Lα conformation is illustrated by the model based on the Phi and Psi angles—as given for the chain structure presented below (Figure 1)

Figure 1A shows the design of a poly-alanine structure with the Phi and Psi angles given in Figure 1B. A flat structure was achieved with sections with a rectilinear propagation, with the present bends obtained through the presence of single (one amino acid in the sequence) amino acids with Rα and Lα conformations, respectively. In the β-structure, all C=O groups are arranged in a direction perpendicular to the chain surface in an anti-parallel pattern. The presence of a single—one amino acid in the sequence with an Rα or Lα conformation causes a bend to appear. These conformations introduce a different system of hydrogen bonds with a local parallel orientation. This is shown in Figure 1C. The system of Phi and Psi angles after transformation |Psi| = |Phi| is presented in Figure 1D, which reveals the linear distribution of the angles |Phi| and |Psi| in the idealised form of a flat chain structure. The whole of this presentation (Figure 1) depicts a model used to determine the degree of order, according to the relation of the Phi and Psi angles in the form |Psi| = |Phi| in the proteins discussed in this paper. These forms are assumed to be present in the actual amyloid structures.

### 2.2. Analysis of the Amyloid Structure Transthyretin

The amyloid form transthyretin (PDB ID-6SDZ) was used as an example of amyloid in a presentation for the analysis carried out in the present study [31].

The maps compiled in Figure 2 reveal differences in the distribution of the Phi and Psi angles on the Ramachandran map in two different forms. The map in Figure 2A shows the complete set of Phi and Psi angles for chain A (red points) (6SDZ), as well as points highlighted in navy blue after eliminating outstanding points (the elimination criterion based on their position on map Figure 2B). The map in Figure 2B shows the same arrangement of the Phi and Psi angles after their transformation into |Psi| and |Phi|. This map reveals a specific distribution of points concentrated on a straight line |Psi| = |Phi|. Therefore, the status of the chains discussed here—components of the amyloid fibrils and the WT form of the proteins in question is assessed using the value of the correlation coefficient CC-0. The higher its value, the greater the share of structuring assumed in the idealised model. The elimination of the points identified as outstanding increases the value of CC (CC-F). The purpose of eliminating outstanding points is, on the one hand, to identify those amino acids which, present in the amyloid fibril, indicate a type of order according to the relation |Psi| = |Phi|, but, on the other, to identify the stabilising factor, as the hydrogen bond with cosα (where α is the angle between the orientation of the C=O group versus the orientation of the N-H group expressed by angle = 180 deg) is the highest. Precisely such a system within the framework of hydrogen bonding is provided by amino acids with a conformation close to |Psi| relation = |Phi|. An additional analysis of the Phi and Psi angles is presented in Figure 3. Interpretation of the angle distribution (Figure 3) reveals relatively long sections with the angles Phi and Psi characteristic of the Beta structure (gray lines), next to which are positions with radically changed R and/or L-helical conformations. Almost no intermediate conformations are present.

A comparison of the Phi and Psi angles in a linear relationship with the amino acid sequence in the transthyretin chain reveals segments featuring the Psi = −Phi conformation. The positions of residues with Lα and Rα conformations are single (one amino acid in the sequence) amino acids between the β-structure segments. It should be noted that the definition of the β-structure, in this case, is the Psi = −Phi conformation.

The set of Phi and Psi angle values in the amyloid and native forms, shown in Figure 3, allows for the evaluation of differences. The positions of the R-helical conformation change. None of the amino acids with this conformation in the native form retained it in the amyloid form. In addition, radical changes are visible—next to segments with a clearly Beta-structural form, there is an amino acid with an R- or L-helical conformation in the amyloid form. In the native form, these changes are gradual. The degree of ordering of the sets of Phi and Psi angles in the amyloid form is much higher compared to the analogous form in the native form. This set of profiles is intended to explain the need to analyse the dispersion of the Phi Psi angles against the idealised system |Psi| = |Phi|, which for transthyretin will be discussed in detail in Section 2.6.

The comparisons given in Figure 1, Figure 2 and Figure 3 visualise the assumptions of the model with an idealised conformation for the form of the flat chain structure present in an amyloid fibril that ensures maximum use of the C=O and H-N groups of peptide bonds in the construction of interchain hydrogen bonds. 

### 2.3. Analysis of the Structure Present in Fibrils in Relation to Their WT Forms

The collective characteristics of the proteins presented in Table 1 summarise the values of the CC-0 correlation coefficient for the structures as they are available in PDB files after eliminating those CC-F residues, which are considered outstanding in relation to the assumed linearity of the relationship |Phi| and |Psi|.

Although the WT form of the ABeta amyloid is not available, these amyloids were also considered in this discussion.

### 2.4. ABeta Amyloids 

No analysis of amyloid structures should overlook the ABeta group amyloid (Figure 4). In this study, three forms with IDs in the PDB database were examined: 2MPZ, 2MVX and 2MXU, respectively. Profiles visualising the values of the Phi and Psi angles reveal segments with an ordering consistent with the model—i.e., segments with Psi = −Phi and clearly marked residues (residues) with Rα or Lα helical conformations, which introduce a turn for β-structure segments with a rectilinear propagation.

The number of residues eliminated (target—increasing the CC value) is small except for 2MVX, although there are segments in the structure of this amyloid with a very distinct linear propagation of the β-structure (Figure 5).

The set of ABeta amyloids seems to support the assumptions of the amyloid fibril chain model.

### 2.5. V Domain of Immunoglobulin Light Chain 

Three elements were analysed: (i) complete chains constituting the domain of light chain V (PDB ID 4BJL), (ii) that part of the domain that corresponds to the fragments present in the amyloid fibril and (iii) the structure of the chains present in the amyloid (PDB ID 6HUD).

A comparison of the Phi and Psi angles (despite the transformation carried out) for the above-mentioned forms reveals significant dispersion and differentiation of the locations of the Phi and Psi angles in the native form of the V domain with no ordering corresponding to the β-structure in the form assumed as Psi = −Phi. Residues eliminated as outstanding in the 3D structure occupy a very scattered location. This mainly concerns locations within the loop, but a few residues present in the β-structure segments also lie in scattered locations.

The dispersion of the Phi and Psi angles (after their transformation) in the amyloid form is characterised by a much higher order compared to the idealised distribution. The elimination of a few residues results in a high CC-F value, which is shown in Figure 6.

The 3D display of the fibril reveals the location of the residues with a conformation deviating from the model at the starting or ending positions of segments with a linear β-structure form.

A comparative analysis of the Phi and Psi angle profiles in the wild type and amyloid forms reveals a significant degree of order, especially in terms of the presence of angles with the relation |Psi| = |Phi| in the amyloid form (Figure 7). Such ordering is not present at all in the native form, except perhaps for the short fragment 15–22 and 85–90. In the amyloid structure, segments with this ordering are present to a much greater extent (8–15, 23–29, 81–95), as are short segments such as, for example, the C-terminal and N-terminal segments of 1–37 and 66–105.

The highlighted items with a pattern of turns consistent with the model in the form of a single (one amino acid in the sequence) Rα- (cyan) and Lα-helical (red) conformation are present in the amyloid form, while the marked analogous conformations in the WY + T form are not very compatible with the assumed model.

A comparative analysis of the Phi and Psi profiles in the WT and amyloid forms reveals one more important feature. The model-oriented sections are present in different sections of the chain in these proteins. Only items 9–11, 18–21 and 86–90 in both forms show similar ordering. These sections, albeit short, can be treated as a kind of “seed” for the propagation expressed by the relationship Psi = −Phi.

### 2.6. Transthyretin

Transthyretin is represented here by four structural forms: three native forms—1DVQ, 1GKO and 1G1O and one amyloid variant—6SDZ. The selection of these structures is as follows: 1DVQ—the reference structure, 1GKO—the form defined as resistant to amyloid transformation [36] and 1G1O—the form defined as aggressively undergoing amyloid transformation [37]. The goal is to determine the possible reasons for this differentiation at the level of the secondary structure.

The arrangement of the Phi and Psi angles in the profile form (Figure 8) enables an identification of the sections that meet the condition Psi = −Phi. It also makes it possible to identify the position of the Rα and Lα helical conformations that result in a turn in the rectilinear propagation of the β-structure segments. To verify the assumptions of the model based on a maximised CC, those residuals treated as outstanding were eliminated, leading to the value CC-F > 0.9 (Figure 9).

The fragments of profiles for Phi and Psi angles meeting this condition are shown in Figure 8B. The residues meeting the assumptions of the model were identified in the 3D presentation of the fibril structure, which reveals the strong presence of an order that conforms to the model, thus confirming its validity. It should be noted that in the sections belonging to CC-F > 0.9, the hydrogen bond system engages each amino acid and groups C=O and H-N in the hydrogen bonding, thereby providing a significant stabilising factor.

The exact distribution of the sections that meet the conditions of the proposed model for the amyloid form are presented in Figure 10. In this figure, three different spatial orientations are presented in order to make the structure more legible. Certain sections are visible that possess features consistent with the model’s assumptions (Figure 10A—blue, Figure 10B—red, Figure 10C—blue sections—outstanding). The lower plot for CC=0.938 and is thus a perfect amyloid form.

In Figure 10B, it is also possible to identify the positions with Rα- and Lα-helical conformations incorporated into the structure of a flat chain form through the introduction of a turn in the chain propagation and which simultaneously engage the C=O and N-H groups of this amino acid in the construction of the hydrogen bond.

An arrangement comprising several residues with Rα and Lα helical conformations is also visible. Such a system maintains the flatness of the structure that is fully engaged in the construction of hydrogen bonds. The various combinations of helical Rα and Lα sets are discussed in detail in [29].

### 2.7. Alpha-Synuclein (aSyn)

The aSyn structure is available in both the native (PDB ID-1XQ8) and amyloid (PDB ID-2N0A) forms. The specificity of the aSyn chain in the amyloid form lies in the engagement of only a 30–100 fragment in the structure of the amyloid fibril. The N- and C-terminal segments have a random coil structure; as a consequence, no chain in its entirety can be taken as representative. Therefore, only fragments 30–100 were analysed for both the WT and the amyloid forms.

As is shown in Figure 11A, an analysis of dependencies in the form |Psi| = |Phi| is not justified due to the complete absence of the β-structure in the WT aSyn form. In the amyloid form of this protein, the presence of the β-structure is significant in that it represents the system |Psi| = |Phi| (Figure 11B,D). Residues with the Rα or Lα helical conformation marked in the 3D structure display reveal localisation in the turns between the β-structure rectilinear segments. The structure of aSyn seems to support the assumptions of the model for identifying the amyloid form in single fibril chains. It is important to note that a parallel system of successive chains in the fibril guarantees the formation of hydrogen bonds in appropriate, identical places, which also guarantees the presence of hydrogen bonds with in the turns.

## 3. Discussion

The flatness of the structure of a single chain, the fibril component, is indisputably a common feature of the fibrillary systems found in amyloids. Proteins with a known WT structure representing a spatial structure (3D) take the form of a flat system (2D) in a fibrillar form. Amyloid transformation from the point of view of chain geometry analysis requires a significant unfolding of the protein in its WT form. This is due to the simple fact of replacing the symmetry operation accompanying the construction of the dimers of the three higher forms of the fourth-order structure as a result of a set of operations of the symmetry axis type (+translation), while only the translation operation is present in the fibril system. This requires an unequivocal shift in the anti-parallel system present in the β-sheet structure in proteins towards a purely parallel structure, which is present in amyloid fibrils.

This is clearly evident in all the examples of proteins whose WT and amyloid structures are available in PDB resources.

This implies the need for a fundamental structural change during amyloid transformation. This is also necessary for another reason. The transthyretin dimer is formed through the construction of two β-sheets constituting their extensions from each monomeric unit. The propagation pattern characterising each β-sheet is a continuation of the antiparallel system. The β-sheet structure in amyloid fibrils is always based in a parallel arrangement (Figure 12). Therefore, the possibility of dimer generation by a protein that transforms into an amyloid cannot be regarded as a “seed” of a fibril. There has been no geometric operation that, without the unfolding of the monomer-derived chain, has given rise to a fibril-based system, where only a translation operation transforms one chain into the other. Dimer formation from an extended β-sheet takes place after the rotation (C2) operation is performed while the antiparallel system in the dimer system is maintained.

The issue of planar/spatial structuralisation in proteins concerns the structuring of hydrogen bonds. The proposed model expressing the relation |Psi| = |Phi| can identify the presence of amino acids that meet this condition. The mutual localisation of amino acids with Rα- and/or Lα-helical conformation is important. The grouping of helical-conformed residues on the longer segment introduces the element of spatiality, which is not the case with amino acid fibrils. The condition is the presence of a single (one position in the sequence) position with this conformation, isolated from the others.

This analysis complements the detailed structural analysis of a-Syn fibrils [40], where the characteristics of the fibrils were considered not from the point of view of the conformation of a single (one residue in the sequence) residue, but rather from the perspective of the third-order structure and the characteristics of these fibrils. The presentation of WT forms in Table 1 should be interpreted with caution. As it was shown in [29], the presence of an ordering of the type |Psi| = |Phi| in WT form cannot be interpreted as a “seed” for the amyloid form. As is shown in [29], neither the Rα- or Lα-helical positions, which play a significant role in the structuring of the amyloid form, occur in the same amino acid positions in their native form. This is also the case with sections that feature a β-structure. The CC-0 values for the native forms of the proteins in question indicate the presence of a suitable system, but this is due to the presence of a β-structure in the plates present in them. This phenomenon is revealed by the profiles of the Phi and Psi angles, where the Rα and Lα-helical positions are marked in both compared forms. These positions for WT and amyloid differ (positions of vertical red lines—Lα and cyan—Rα are indicated for different amino acids).

The choice of transthyretin representatives in the forms of 1GKO, 1DVQ and 1G1O was dictated by different characteristics found in laboratory tests. 1DVQ is treated as the reference form here. On the other hand, 1GKO has been observed in experimental conditions to be resistant to amyloid transformation [36], while it has been observed that 1G1O has undergone this transformation aggressively [37].

## 4. Materials and Methods

### 4.1. Data 

The proteins analysed in the current work are presented in Table 2. 

The list of proteins is limited to those whose native and amyloid forms were available at the time of this work (June 2022). As the number of such sets of proteins increases, the analysis will continue.

### 4.2. Model—Description 

The model presented in this study, which expresses the structural specificity of amyloid fibrils, is based on the assumptions for the geometric representation of polypeptide structuring expressed by the radius of curvature (Ln (R) where R—radius of curvature) and the angle expressing the mutual orientation of the adjacent planes of peptide bonds. This angle expresses the amount of rotation on the virtual Cα-Cα bond. This angle for the helix is 0 deg, while the value of that angle for the β-structure is given as V-angle = 180. Changing this angle results in a change in the radius of the curvature from the low value known for α-helix to the theoretically infinitely large radius of curvature for the β-structure. All intermediate structures are expressed by a suitable angle 0 and lt; V and lt; 180 deg and a suitable radius of curvature. This relationship is illustrated by the graph (Figure 13.).

The distribution of the radius of curvature on the Ln (R) scale and the size of the V angle are shown in Figure 14, respectively. Visible are areas with low values of both R and V as well as areas where these values show a rapid increase. The maximum values of V and R are achieved in the area, which satisfies the condition of the dependence Psi = −Phi.

The specificity of amyloid structuring is expressed by the appropriate system of Psi = −Phi type conformations, which generates a system in rectilinearly propagating a β-structure. The turns disrupting this rectilinear propagation are due to the presence of single residues with a conformation corresponding to Rα- or Lα-helicals.

The specific system of hydrogen bonds, perpendicular to the plane of the polypeptide chain as observed in amyloid fibril structures, is preserved for both of the given conformations. Only the system of hydrogen bonding changes from an antiparallel form in the β-structure to a locally parallel system in the amino acid with an Rα- or Lα-helical conformation. Such a system is ideally realised for the helical conformations, but also for the angles surrounding these positions. This is made possible by the broad area on the Ramachandran map with low values for both R- and V-angles (Figure 15). On the other hand, for the β-structural conformation, a change in angle beyond the line Psi = −Phi results in a fairly significant change in the radius of curvature, which may result in the introduction of spatiality in the system, which is not present in amyloid forms.

## 5. Conclusions

The polypeptide chain structure in the amyloid fibril comprises two features: a flat structure along the entire length of the chain and a specific arrangement of the hydrogen bond network that runs perpendicular to the plane of the chain. Both these conditions are observed in amino acids with a conformation satisfying the condition |Psi| = |Phi|. This condition is met by amino acids with the β-strand conformation, together with the additional assumption of a Psi = −Phi relation. Such a system generates a rectilinear system with endless propagation. The presence of turns is the result of single (one amino acid in the sequence) conformations with angles characteristic of R and L Rα and Lα-helices. The juxtaposition of these two conformational characteristics leads to the definition of the model expressed by the relation |Psi| = |Phi| for structuring the amyloid fibril. An additional feature of amyloid fibrils is the parallel system for the mutual positioning of adjacent chains. This system distinguishes the structural form present in the fibril in relation to globular proteins, where the network of hydrogen bonds is formed β-sheets with an anti-parallel system.

As a consequence, the presence of highly ordered conformations meeting the defined condition in globular proteins cannot be regarded as the presence of a “seed” for amyloid formation. Amyloid transformation requires a significant degree of unfolding to obtain a β-sheet parallel system from the primary anti-parallel system in native proteins.

Furthermore, the structural dependence on the environment known from experimental studies in relation to the amyloid transformation [14,15,16,17,18,19,20,21,22,40] justifies the observations presented in [41].

The aim of the study was to quantitatively evaluate the structural changes of amyloid forms in relation to wild type forms based on the model proposed for the description of amyloid structures. This model expressed by the relation |Psi| = |Phi| results from the observations of the presence of the Beta structure with the Psi = −Phi specificity, ensuring the straight-line propagation of the chain in combination with the single residues of the Rα or Lα conformation. The proposed relationship takes into account both the specificity of the Beta-structure and helical forms.

A significant approximation of the conformation in amyloid forms to the native forms expressed using the proposed model was demonstrated. The model expressing the geometry of the polypeptide chain using a V-angle and a radius of curvature R allows the expression of the Beta-structure as a form with a large value of the curvature radius with the orientation of the peptide bond planes with a V-angle = 180 deg. The geometry of segments with single residues in the helical conformation is expressed in this model by a small radius of curvature R (change of rectilinear propagation due to twisting) with V-angle = 0 deg. This arrangement preserves the orientation of the C=O and H-N groups in the arrangement perpendicular to the flat plane of the chain (a form typical for polypeptide chains present in fibrils), guaranteeing the possibility of building a network of hydrogen bonds involving almost all amino acids, which is not observed in the case of Beta-sheets present in the native protein forms.

## Figures and Tables

**Figure 1 ijms-24-00154-f001:**
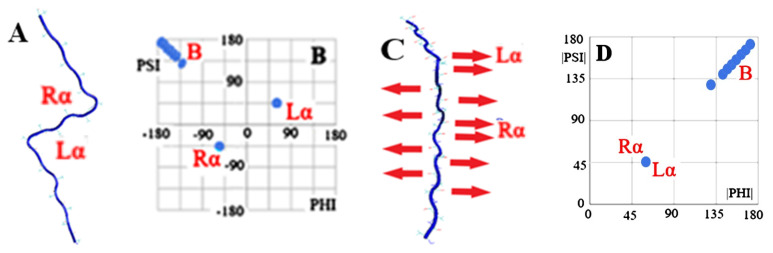
An example of a poly-alanine (**A**) structure with Phi and Psi angles (**B**) with hydrogen bond turns highlighted (a representation of the C=O group orientation) (**C**) with the Phi and Psi angles after transformation |Phi| = |Psi| (**D**).

**Figure 2 ijms-24-00154-f002:**
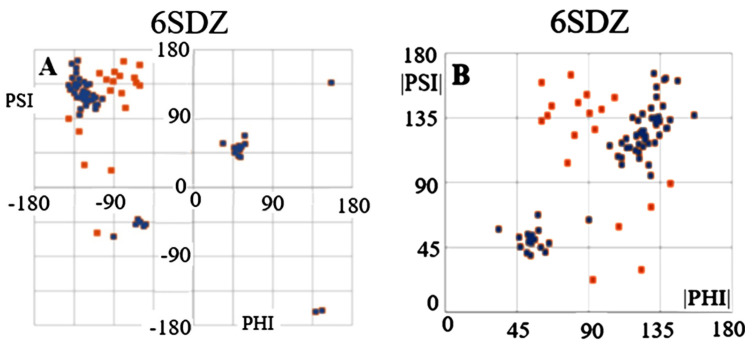
Ramachandran map: (**A**)—a comparison of the Phi and Psi angles for the complete chain of the amyloid form transthyretin (6SDZ), navy blue—points treated as outstanding; (**B**)—processed Phi and Psi angles—absolute values of angles highlighted as navy blue points—set after eliminating outstanding positions.

**Figure 3 ijms-24-00154-f003:**
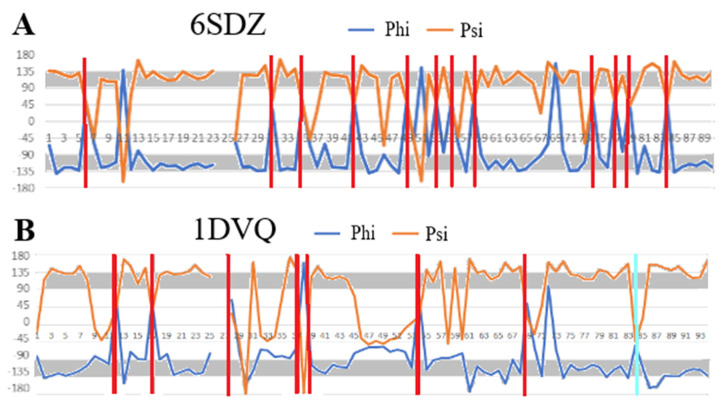
A comparison of the Phi and Psi angles for the (**A**)—amyloid (6SDZ) chain and its (**B**)—wild type form of transthyretin (1DVQ). Red diagram—Phi angles, blue diagram—Psi angles. Gray lines—distinguishing a β-structure conformation with the relation Psi = −Phi. Vertical lines: red—amino acid positions with Lα conformation, cyan lines—amino acid positions with Rα conformation. The shaded lines are to visualise the relation between Phi and Psi which is significantly more regular in amyloid form.

**Figure 4 ijms-24-00154-f004:**
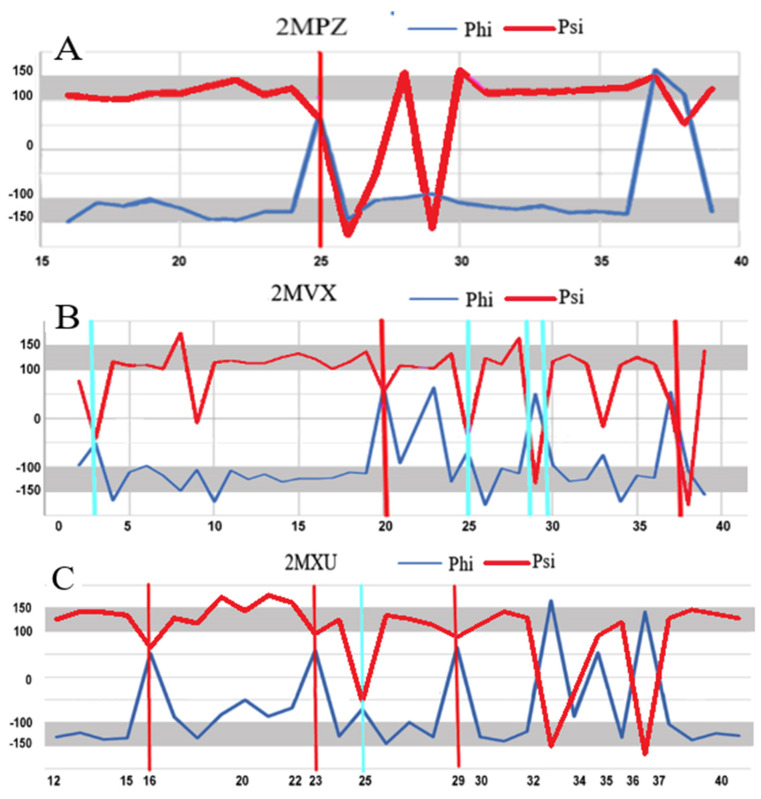
Comparisons of Phi and Psi angle profiles in the ABeta amyloids: (**A**)—2MPZ; (**B**)—2MVX; (**C**)—2MXU. Gray area in (**A**)—highlighting areas for Phi and Psi with the relation Psi = −Phi. Vertical red lines—amino acid positions with Lα helical conformation; cyan—Rα helical.

**Figure 5 ijms-24-00154-f005:**
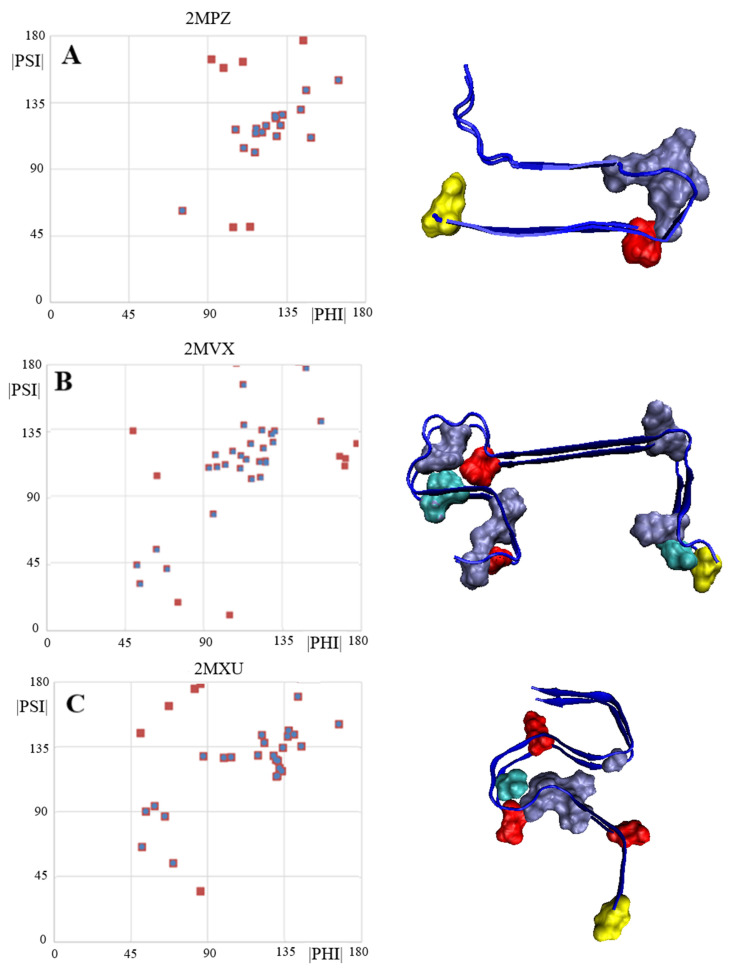
List of point distributions representing Phi and Psi angles following transformation |Psi| and |Phi|, in the case of: (**A**)—2MPZ—Amino acids were highlighted in the 3D presentation as in point B; (**B**)—2MVX—in a 3D presentation ice blue—items eliminated as outstanding; red—amino acids with Lα conformation; cyan—amino acids with Rα conformation; (**C**)—2MXU—Amino acids were highlighted in the 3D presentation as in point B. Red points on the maps represent all conformations present in the polypeptide, blue points—points remaining after outstanding points are eliminated. 3D presentations—yellow residues—N-terminal of amino acids for easy navigation.

**Figure 6 ijms-24-00154-f006:**
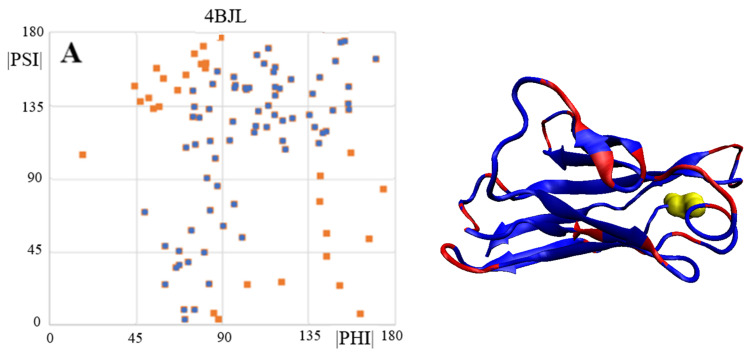
The 3D display of the fibril reveals the location of residues with a conformation deviating from the model at the starting or ending positions of segments with a linear β-structure form. (**A**) – the complete V domain of IgG. (**B**)—part of domain V of the IgG light chain as present in amyloid form of this amyloid—highlighted in the 3D presentation as in point A. (**C**)—amyloid form of domain V of IgG light chain-amino acids highlighted in a 3D presentation as in Figure 5. *—denotes the section of V domain present in the amyloid form.

**Figure 7 ijms-24-00154-f007:**
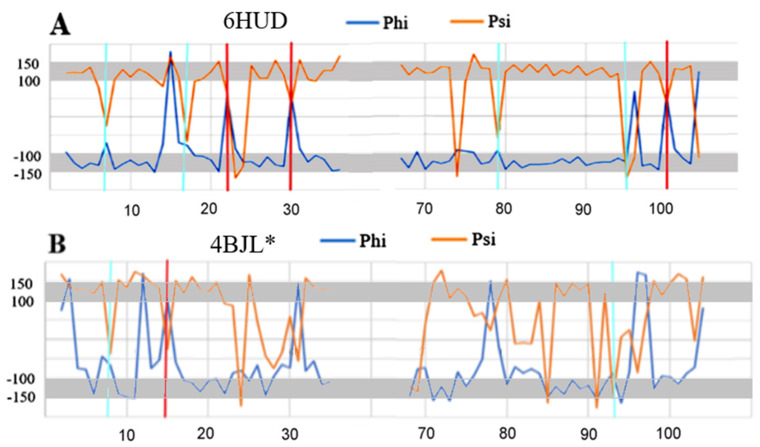
List of profiles for the Phi and Psi angles in: (**A**)—amyloid form of the V domain of IgG light chain (**B**)—native form of the V domain of IgG light chain for the segments as present in the amyloid form. Vertical lines: red—helical Lα conformation, cyan—helical Rα conformation. “*” denotes the part of chain as present in amyloid form.

**Figure 8 ijms-24-00154-f008:**
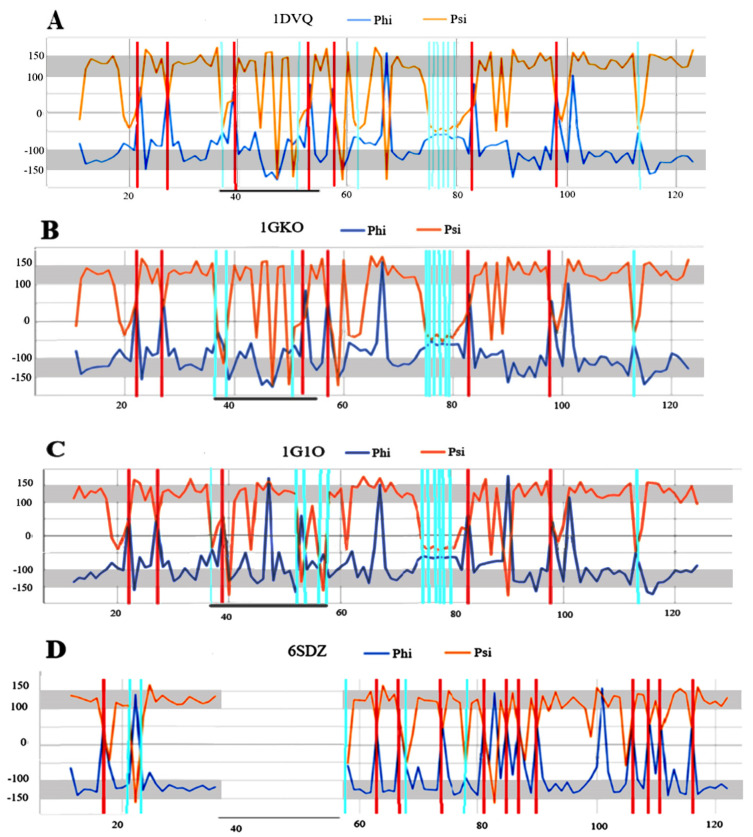
Summary of profiles for the Phi and Psi angles in the transthyretins: (**A**)—1DVQ—native form—reference (**B**)—1GKO—form resistant to amyloid transformation; (**C**)—1G1O—form that undergoes aggressive amyloid transformation (**D**)—6SDZ—amyloid form of transthyretin; Vertical lines—cyan—positions with Rα conformation; red—helical Lα. Horizontal; grey bottom line on (**A**–**C**)—section of chain absent in the amyloid form.

**Figure 9 ijms-24-00154-f009:**
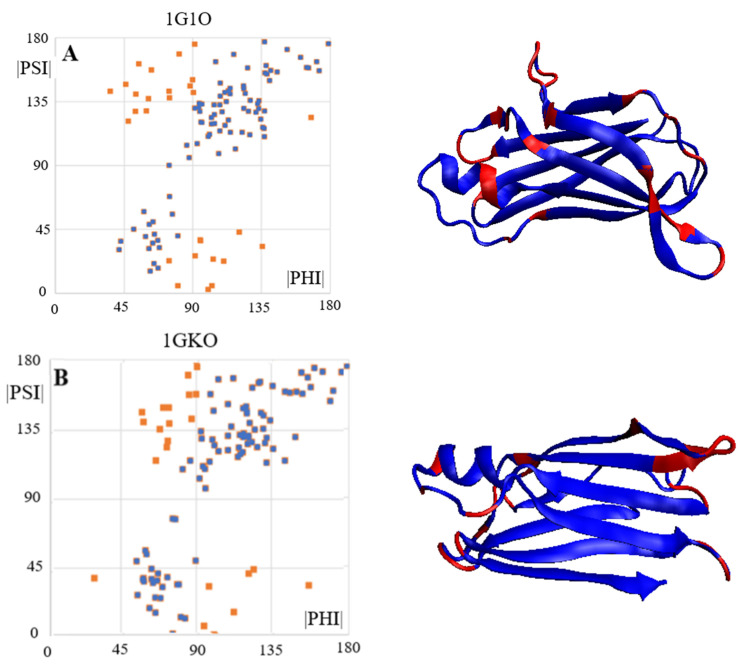
Location of points representing the Phi and Psi angles for: (**A**)—1G1O and the distribution of items treated as outstanding (red in the 3D presentation); (**B**)—1GKO and the distribution of items treated as outstanding (red in the 3D presentation); (**C**)—1DVQ. (**D**)—1DVQ. * Colour scheme as in previous figures.

**Figure 10 ijms-24-00154-f010:**
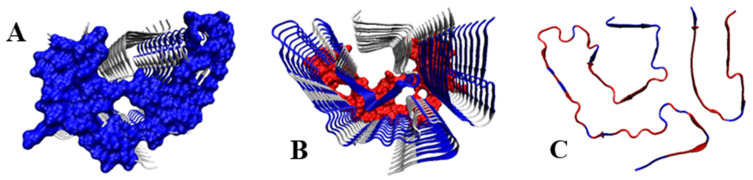
Presentation of a 3D transthyretin structure. (**A**)—sections (navy blue) that meet the model’s conditions are highlighted, sections of proteins—amino acids with a conformation treated as outstanding; (**B**)—red sections are sections that meet the conditions of the assumed model, sections of proteins—amino acids with a conformation treated as outstanding. Navy blue segments in the remaining chains—conformations meet the conditions of the model; (**C**)—single chain: navy blue sections—sections with linear propagation, red sections—turns performed by a single (one amino acid in the sequence) Lα or Rα helical conformation (or a detailed overview in [29]).

**Figure 11 ijms-24-00154-f011:**
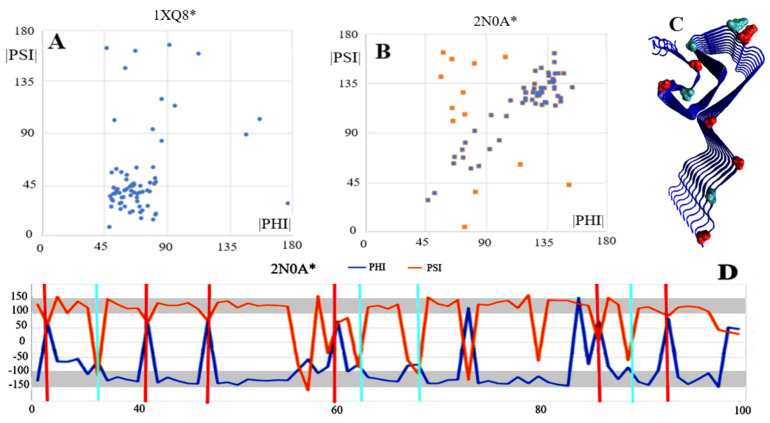
Characteristics of aSyn in the form of: (**A**)—distribution of angles |Phi| and |Psi| in native form (1XQ8*); (**B**)—distribution of angles |Phi| and |Psi| in amyloid form (2N0A*); (**C**)—3D form of amyloid with highlighted residues: cyan—Rα conformation, red—helical Lα conformation, as these positions were highlighted in the profiles in D. (**D**)—set of Phi and Psi profiles for 2N0A* with highlighted positions Lα—red and Rα helical—cyan.

**Figure 12 ijms-24-00154-f012:**
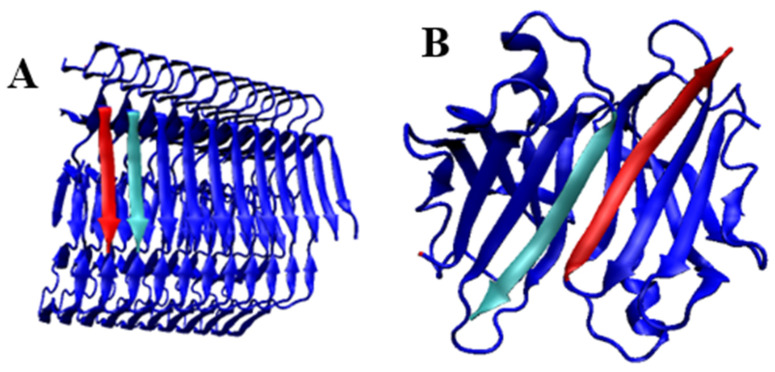
Arrangement of two β-strands in the β-sheet layout—the same chain fragments distinguished by colour—transthyretin structure. (**A**)—parallel in fibril (6SDZ); (**B**)—antiparallel in WT form dimer (1DVQ).

**Figure 13 ijms-24-00154-f013:**
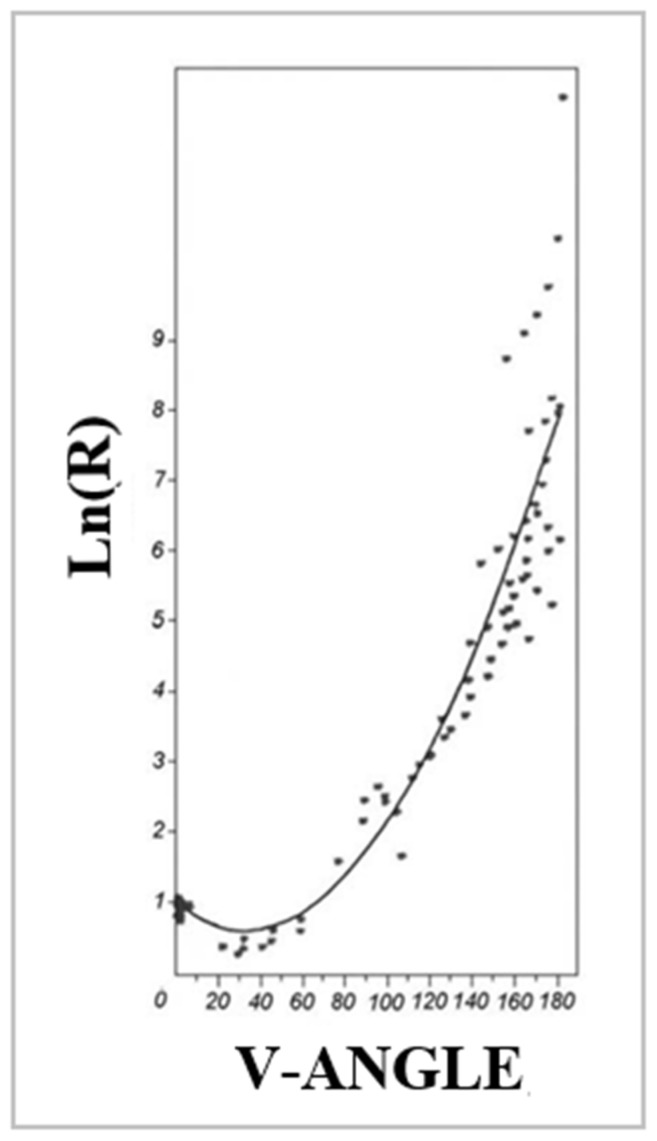
Basis for the model generating an early intermediate structure using an analysis of geometric parameters–dependence of R (logarithm scale) on the V-angle—a parabolic curve was determined by approximating the points representing low-energy areas on the Ramachandran map.

**Figure 14 ijms-24-00154-f014:**
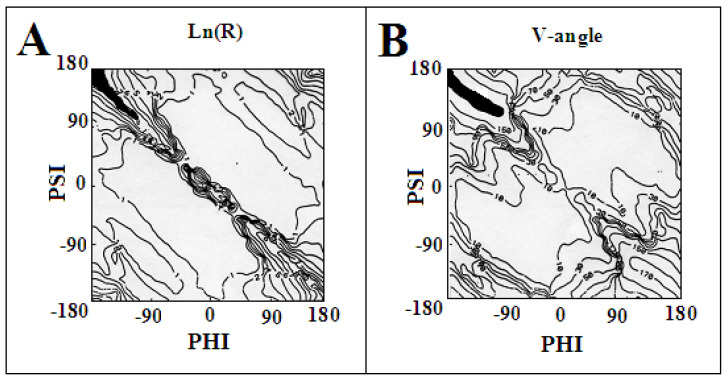
Distribution of the values of geometric parameters describing the polypeptide arrangement: (**A**)—radius of curvature on the Ln (R) scale; (**B**)—V-angle—angle expressing the relative orientation of the planes of peptide bonds on the virtual Cα-Cα bond; The maps highlight (black) areas with the highest V-angle values and the highest values of the curvature radius.

**Figure 15 ijms-24-00154-f015:**
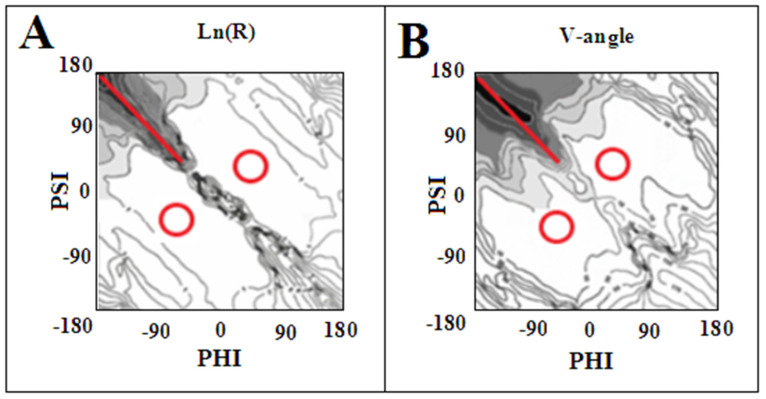
Maps of the distribution of geometric parameters describing the structure of the polypeptide: (**A**)—radius of curvature on the Ln (R) scale; (**B**)—V-angle—the size of the rotation angle areas that meet the conditions |Psi| = |Phi| are highlighted with a red line and red circles.

**Table 1 ijms-24-00154-t001:** Summary of CC-0 and CC-F values for the discussed proteins. Also given are the positions of amino acids considered outstanding and the elimination of which from the CC calculations results in an increase in its value.

PDB ID	CC-0	RESIDUES ELIMINATED	CC-F
2MPZ [32]	0.364	**26**, 27, 28, **29**, 30	0.860
2MXU [33]	0.403	19, 20, 21, 22, **34**	0.820
2MVX [34]	0.521	4, 9, 10, 23, **26**, **29**, 33, **34**, 38	0.860
1DVQ [35]	0.619	21, **24**, 29, 35, 36, 43, 44, 49, 50, 52, 65, 66, 82, **86**, 88, **100**, **102**, 114	0.862
1GKO [36]	0.570	20, 21, 24, 26, 29, 35–39, 43, 44, 49, 50, 52, 65, 66, 82, 85, 88, 100, 102, 114	0.852
1G1O [37]	0.477	19–21, **24**, 26, 29, 35, 36, 38, 43, 44, 49, 52–54, 56, 57, 62, 64, 66, 82, 88, 89, **100**, **102**, 114	0.855
1DVQ * [35]	0.663	11, 19–21, **24**, 26, 29, 65, 66, 82, **86**, 88, 96, **100**, 101, **102**, 114	0.913
1GKO * [36]	0.657	**12**, 19–21, **24**, 26, 29, 65, 66, 82, 85–88, **93**, 100–102, 117	0.918
1G1O * [36]	0.508	19–21, **24**, 26, 29, 62, 64, 66, 82, **86**, 88, **100**, **102**, 114	0.842
6SDZ [31]	0.622	**12**, 23, **24**, 69, 70, 75, 84, **86**, 91, **93**, 99, **100**, **102**, 105, 110, 112, 113	0.938
4BJL-V [38]	0.108	7, 14, 15, **16**, 22, 24, 27, **31**–33, 41, 42, 44, 45, 51, 53, 55, 56, 67, 73, 78, 79, 82, 94, **96**, 97	0.635
4BJL-V * [38]	0.120	2–4, 7, **13**, 14, 15, **16**, 22, 24, 27, **31**–33, 67, 69, 73, 77–79, 94, **96**, 97	0.694
6HUD [39]	0.495	**13**, **16**, 23, **31**, 74, 76, **96**, 101	0.730
1XQ8 * [23]	0.256		
2N0A * [24]	0.512	32–34, 36, 57–59, 68, 80, 85–87, 98	0.913

* Only the amyloid fraction is analysed. Numbers given as bold—the same items eliminated in both native and the amyloid form of the protein in question.

**Table 2 ijms-24-00154-t002:** The proteins analysed in the current work (amyloid forms in bold).

PDB ID	Ref.
2MPZ [32]	[32]
2MXU [33]	[33]
2MVX [34]	[34]
1DVQ [35]	[35]
1GKO [36]	[36]
1G1O [37]	[37]
**6SDZ [31]**	**[31]**
4BJL-V [38]	[38]
**6HUD [39]**	**[39]**
1XQ8 [23]	[23]
**2N0A [24]**	**[24]**

## Data Availability

All data can be available on request addressed to corresponding author.

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
