# Peer review of "Secondary Structure in Amyloids in Relation to Their Wild Type Forms"

_ijms, 2022, doi:10.3390/ijms24010154_

Round 1

Reviewer 1 Report

Roterman et al. compare the amyloid form and globular froms of proteins in terms of protein backbone dihedrals and a new V-variable. They indicate the presence of the | Psi | = | Phi | condition as characteristic for amyloid structures in order to generate flat conformations. This condtions is automatically fulfilled for beta-strands (which is the most common conformation in amyloids) and makes the Ralpha and Lalpha conformations equivalent (which are common in turns in amyloids. The authors find much less order in the globular forms of the proteins. I have a few comments and concerns:

1. It is not really clear to me what one learns from this type of analysis beyond what is already present in the Ramachandran plots for the globular and amyloid forms. The authors need to explain better the take home message of the paper.

2. The paper is very descriptive and gives no explanation as to why proteins adopt an amyloid form with a flat polypeptide structure. What is the physical reason for the observations made by the authors. Can the authors use the analysis to make any kind of prediction of amyloid formation/structure?

3. From the plots of phi and psi values along a chain (several in the paper) one gets the impression that phi and psi undergo at some positions drastic changes because they change from something near +160 to -160. However, this is mealeading because the realy change is only 40 degrees due to the periodicity of the angle. To avoid this problem it is better to set the cut for the periodicity not at 180/-180 but at a different value to avoid these "artificial" jumps in the plots (or to change to other types of plos, e.g circular plots for the dihedral angles).

4. I do not see an advantage of working with the V-angle instead of phi/psi. The authors need to explain what additional information is provided by using the V-variable compared to the phi/psi variables.

5. What is the wildtype form of a protein (is it the globular 3D fold, this should be somewhere explained..)?

Author Response

REVIEWER I

Open Review

English language and style

( ) English very difficult to understand/incomprehensible
( ) Extensive editing of English language and style required
( ) Moderate English changes required
(x) English language and style are fine/minor spell check required
( ) I don't feel qualified to judge about the English language and style

Yes

Can be improved

Must be improved

Not applicable

Does the introduction provide sufficient background and include all relevant references?

(x)

( )

( )

( )

Are all the cited references relevant to the research?

(x)

( )

( )

( )

Is the research design appropriate?

( )

(x)

( )

( )

Are the methods adequately described?

(x)

( )

( )

( )

Are the results clearly presented?

( )

(x)

( )

( )

Are the conclusions supported by the results?

( )

(x)

( )

( )

Comments and Suggestions for Authors

Roterman et al. compare the amyloid form and globular froms of proteins in terms of protein backbone dihedrals and a new V-variable. They indicate the presence of the | Psi | = | Phi | condition as characteristic for amyloid structures in order to generate flat conformations. This condtions is automatically fulfilled for beta-strands (which is the most common conformation in amyloids) and makes the Ralpha and Lalpha conformations equivalent (which are common in turns in amyloids. The authors find much less order in the globular forms of the proteins. I have a few comments and concerns:

  1. It is not really clear to me what one learns from this type of analysis beyond what is already present in the Ramachandran plots for the globular and amyloid forms. The authors need to explain better the take home message of the paper.

The aim of the study is to determine the differences in the occurrence of Phi and Psi angles in amyloid in comparison with the native structures of these proteins. These differences are quantified by showing that the clustering of the positions of the Phi and Psi angles in amyloids is closer to the relation |Psi|=|Phi|. Thus, a model was proposed to determine the criterion for evaluating the restoration of the idealized form present in amyloids. Obtaining a flat structure propagating linearly for the Beta-structure form is obvious. The combination of angles satisfying the condition Psi = - Phi guarantees rectilinear propagation of the chain. In addition, such a set of conformations results in the orientation of the C=O and H-N groups perpendicular to the chain plane. This orientation of these groups is a preparation for interactions mediated by hydrogen bonds between chains from adjacent fibril layers.

Straight line propagation cannot continue indefinitely. The introduction of twists (turns) in chain propagation is achieved by means of single R- or L-helical amino acids (right- and left-handed helix). The R- and L-helical conformation also provides a system with the C=O and H-N groups perpendicularly oriented to the chain plane. Therefore, it is possible to continue the system of hydrogen bonds in the perpendicular to the chain plane. The only difference resulting from the presence of a single R- or L-helical conformation is the change in the direction of the hydrogen bond (if this bond is oriented forming a  dipole). The continued Beta-structural arrangement introduces an anti-parallel arrangement of hydrogen bonds, while a single residue with an R- or L-helical conformation introduces a locally parallel arrangement.

Hydrogen bonds are the stabilizer of the Beta structure in globular proteins. The difference is mainly in the resulting flat chain structure - the fibril component. This is due to the fact that the continuation of the propagation of the Beta structure in globular proteins proceeds gradually by means of loop segments. In amyloids, the change in the direction of propagation takes place in strictly defined positions, in most cases carried out by means of one amino acid of the R- and L-helical conformation.

The analysis of the location of the Phi and Psi angles in amyloids in many cases has the form of the Psi = - Phi line. Globular proteins represent a much larger dispersion of the locations of these angles on the Ramachandran map.

The analysis of the distribution of the Phi and Pasi angles on the Ramachandran map is an obvious visualization of the position of these angles. The model proposed in the paper, however, introduces a kind of model expressing the criterion for evaluating the degree of order present in amyloids. The work showed that the ordering |Psi|=|Phi| this is the criterion. This is, of course, a mathematical procedure, but this procedure makes it possible to quantify the contribution of this type of arrangement to any molecule, including proteins in their WT and amyloid forms.

  1. The paper is very descriptive and gives no explanation as to why proteins adopt an amyloid form with a flat polypeptide structure. What is the physical reason for the observations made by the authors. Can the authors use the analysis to make any kind of prediction of amyloid formation/structure?\

The answer to the question why amyloid proteins adopt this type of conformation is not the subject of analysis. The commonly accepted interpretation is based on environmental considerations. In the current work, the aim is to determine the specificity of the amyloid structure expressed by the idealized relation |Psi|-|Phi|| as characteristic of amayloid forms.

  1. From the plots of phi and psi values along a chain (several in the paper) one gets the impression that phi and psi undergo at some positions drastic changes because they change from something near +160 to -160. However, this is mealeading because the realy change is only 40 degrees due to the periodicity of the angle. To avoid this problem it is better to set the cut for the periodicity not at 180/-180 but at a different value to avoid these "artificial" jumps in the plots (or to change to other types of plos, e.g circular plots for the dihedral angles).

  1. I do not see an advantage of working with the V-angle instead of phi/psi. The authors need to explain what additional information is provided by using the V-variable compared to the phi/psi variables.

The main advantage of using the V-angle and radius R of curvature parameters is the ease of visualizing the structure expressed by these parameters. Angles Phi, Psi, apart from the standard secondary-structure, do not give the possibility of imagining the corresponding structure. If a sequence of V and R parameter values is given for the structure, the shape of the chain can be (approximately) reconstructed.

The introduction of the V-angle is aimed at tracking changes in the mutual position of the peptide bond planes. Such a description of the structure was used in publication:

Hayward S. Peptide-plane flipping in proteins. Protein Sci. 2001 ; 10(11): 2219–2227. doi: 10.1110/ps.23101,

where the authors use the term peptide plane flapping. It has just been shown that the change in the mutual orientation of the peptide bond planes is characteristic of the amyloid system, where the change in the mutual orientation of the planes is radical. In the proposed model, the V-angle just expresses quantitatively the change in the mutual orientation of the peptide bond planes.

  1. What is the wildtype form of a protein (is it the globular 3D fold, this should be somewhere explained..)?

This term has been explained - WT - means wild type - the structure that the protein represents in normal (physiological) conditions showing biological activity. The WT structure excludes any experimental interference. An amyloid structure that loses biological activity is not a WT structure because it is the result of a process called misfolding.

Reviewer 2 Report

The manuscript by Roterman and collaborators deals with the comparison of the secondary structure of protein amyloids in relation to their wild-type conformation, using a number of proteins with their respective PDBs for the wild-type and amyloid forms. Although the thematic is of interest to the protein folding and protein misfolding field, the text is hard to follow and there are a group of misinterpretations that need to be corrected/modified.

Beginning with the title, the authors should write WT in full.

The abstract is too vague, there is no introduction in it, the research hypothesis is not presented, nor the methodology. It is very difficult to follow both the idea of the analysis and the novelty of the results.

Examples of incorrections:

Line 33: ‘as they disturb the functioning of the nervous system’. Amyloidogenic diseases occur in other organs and tissues, besides the nervous system. The authors inclusive evaluate proteins not related to neurodegenerative diseases in this paper.

This is a misinformation: ‘The main area of research from the perspective of treatment is gene therapy, in particular the use of stem cells.’

Lines 56-57: Phi and 56 Psi angles in amino acids: this is wrong, dihedral angles are found along the primary structure and not in free amino acids.

Lines 55-80: difficult to follow the different orientations without an explaining figure/scheme.

Lines 65-66: ‘The turns in the chain arrangement are derived from single amino acids with an Rα- or Lα-helical conformation.’ The authors should write in full and explain these orientations.

The authors confuse the protein with the amyloid structure adopted by the protein.

The quality of several figures is low.

There is no detailed description of the V angle in the introduction.

In general, I found the manuscript hard to read and did not find the novelty of the structural analysis based on the Ramachandran plot and the V angle. Besides, more details are needed regarding the chosen PDB files. Most of the high-resolution structures for protein amyloid deposited in the PDB are from isolated domains of each protein. In addition, the amyloid conformation shown in a PDB is characteristic of a type of amyloid, as several amyloid structures are polymorphic. Therefore, there is a lack of background on how each amyloid was obtained and how is this domain comparable to the full-length wild-type protein.

Author Response

REVIEWER II

Open Review

English language and style

( ) English very difficult to understand/incomprehensible
( ) Extensive editing of English language and style required
(x) Moderate English changes required
( ) English language and style are fine/minor spell check required
( ) I don't feel qualified to judge about the English language and style

Yes

Can be improved

Must be improved

Not applicable

Does the introduction provide sufficient background and include all relevant references?

( )

(x)

( )

( )

Are all the cited references relevant to the research?

(x)

( )

( )

( )

Is the research design appropriate?

( )

(x)

( )

( )

Are the methods adequately described?

( )

(x)

( )

( )

Are the results clearly presented?

( )

(x)

( )

( )

Are the conclusions supported by the results?

( )

(x)

( )

( )

Comments and Suggestions for Authors

The manuscript by Roterman and collaborators deals with the comparison of the secondary structure of protein amyloids in relation to their wild-type conformation, using a number of proteins with their respective PDBs for the wild-type and amyloid forms. Although the thematic is of interest to the protein folding and protein misfolding field, the text is hard to follow and there are a group of misinterpretations that need to be corrected/modified.

Beginning with the title, the authors should write WT in full.

CORRECTED

The abstract is too vague, there is no introduction in it, the research hypothesis is not presented, nor the methodology. It is very difficult to follow both the idea of the analysis and the novelty of the results.

Abstract changed.

Examples of incorrections:

Line 33: ‘as they disturb the functioning of the nervous system’. Amyloidogenic diseases occur in other organs and tissues, besides the nervous system. The authors inclusive evaluate proteins not related to neurodegenerative diseases in this paper.

This statement has been extended to include the phenomenon of amyloidosis occurring in other organs like kidney or  heart.

This is a misinformation: ‘The main area of research from the perspective of treatment is gene therapy, in particular the use of stem cells.’

This statement is taken from other publications on amyloidosis from the point of view of the therapy used.

Lines 56-57: Phi and 56 Psi angles in amino acids: this is wrong, dihedral angles are found along the primary structure and not in free amino acids.

It has been corrected – „single amino acid in a sequence”

Lines 55-80: difficult to follow the different orientations without an explaining figure/scheme.

Lines 65-66: ‘The turns in the chain arrangement are derived from single amino acids with an Rα- or Lα-helical conformation.’ The authors should write in full and explain these orientations.

Figure 1 has been placed together with the fragment indicated as requiring explanation in graphic form. The linearly propagating chain representing the Beta structure bends at the amino acid site where the R- or L-helical conformation appears. We are talking here not about "isolated" but about a "single" amino acid in a sequence of amino acids in a chain.

The authors confuse the protein with the amyloid structure adopted by the protein.

Added clarification to distinguish amyloid forms - term - "amyloid"

The non-amyloid form is referred to as the "native form of the protein"

The quality of several figures is low.

Corrected

There is no detailed description of the V angle in the introduction.

Added

 In general, I found the manuscript hard to read and did not find the novelty of the structural analysis based on the Ramachandran plot and the V angle.

Added relevant new fragment  at the end of Conclusions.

Besides, more details are needed regarding the chosen PDB files. Most of the high-resolution structures for protein amyloid deposited in the PDB are from isolated domains of each protein. In addition, the amyloid conformation shown in a PDB is characteristic of a type of amyloid, as several amyloid structures are polymorphic. Therefore, there is a lack of background on how each amyloid was obtained and how is this domain comparable to the full-length wild-type protein.

The phenomenon of polymorphism is discussed in a separate paper analyzing the numerous available forms of Alpha-Synuclein amyloids. In the current work, it is limited to 2N0A, because only this structure represents a complete chain (140 aa) (current status of this work - preliminary accepted). The choice of structures present in the analysis results from the availability of both native and amyloid forms for the same protein. On the day of the analysis, only the proteins present in the paper were available in both structural forms in the PDB.

Round 2

Reviewer 1 Report

I carefully read th revised version of the manuscript and the response to my concerns. I think the authors respeonded successfully to my concerns. The additions and corrections are very helpful for the reader to understand the main message of the paper.

Reviewer 2 Report

The authors have answered the main comments raised by this reviewer and modified the text accordingly. However, to improve readability, I recommend that the manuscript is checked for language and grammar.